# Integrating the Gut Microbiome and Stress-Diathesis to Explore Post-Trauma Recovery: An Updated Model

**DOI:** 10.3390/pathogens11070716

**Published:** 2022-06-23

**Authors:** Manasi Murthy Mittinty, Joshua Y. Lee, David M. Walton, Emad M. El-Omar, James M. Elliott

**Affiliations:** 1Faculty of Medicine and Health, University of Sydney, Sydney, NSW 2065, Australia; 2School of Physical Therapy, Western University, London, ON N6G 1H1, Canada; jlee2793@uwo.ca (J.Y.L.); dwalton5@uwo.ca (D.M.W.); 3UNSW Microbiome Research Centre, St George and Sutherland Clinical Campuses, School of Clinical Medicine, Faculty of Medicine and Health, University of New South Wales, Sydney, NSW 2052, Australia; e.el-omar@unsw.edu.au; 4School of Health Sciences, Faculty of Medicine and Health, The Kolling Institute, The University of Sydney, Sydney, NSW 2065, Australia; jim.elliott@sydney.edu.au; 5The Northern Sydney Local Health District, Sydney, NSW 2006, Australia; 6Department of Physical Therapy and Human Movement Sciences, Feinberg School of Medicine, Northwestern University, Chicago, IL 60611, USA

**Keywords:** musculoskeletal conditions, gut microbiome, gut flora, traumatic injury, stress

## Abstract

Musculoskeletal conditions of traumatic and non-traumatic origin represent an ongoing health challenge. While the last three decades have seen significant advancement in our understanding of musculoskeletal conditions, the mechanisms of a delayed or lack of recovery are still a mystery. Here, we present an expansion of the integrated stress-diathesis model through the inclusion of the gut microbiome. Connecting the microbiome with known adverse neurobiologic, microbiologic and pathophysiologic sequelae following an injury, trauma or stressful event may help improve our knowledge of the pathogenesis of poor recovery. Such knowledge could provide a foundation for the exploration and development of more effective interventions to prevent the transition from acute to chronic pain.

## 1. Introduction

Nearly one-third of the world’s population (~2.5 billion people) will experience a non-catastrophic musculoskeletal (MSK) injury at some point in their life [1]. While more than 50% of those injured should expect to recover within the first 6–12 weeks, about 20–25% will transition to chronic pain, experiencing reductions in physical activity, impaired mental health, higher healthcare costs, and poor quality of life [2,3,4]. Rightfully, the Global Burden of Diseases [5] identified MSK conditions as the second highest contributors to years lived with disability (YLDs), with a ~60% rise in disability-adjusted life years (DALYs) from 1990 to 2015. Unfortunately, nearly 30 years of research into the prevention of, and interventions for, MSK conditions and pain have done very little to reduce their relative burden. A potential reason for this remains the heterogeneity of the clinical signs and symptoms that are likely influenced by pre-existing resiliencies and/or vulnerabilities and the socioenvironmental context within which the individual functions [6,7,8].

Central to this thesis, a diathesis-stress model has been proposed using a common clinical musculoskeletal condition triggered by non-catastrophic motor vehicle accidents, whiplash-associated disorder (WAD), to scaffold the theoretical and empirical mechanistic pathways [8]. The model posited that in order to achieve a better understanding of patient-centred outcomes, a multisystem interaction—integrating psycho- and neurobiological processes—was needed. It was in this spirit that the model was presented but not finalized, recognizing that new evidence, pathways, and interactions would emerge and likely propel the model and the field forward. 

Based on emerging evidence documenting a link between the microbiome and multiple chronic conditions, the gut microbiome warrants inclusion in the existing diathesis-stress model [8]. Some of the key conditions exhibiting this link include irritable bowel syndrome (IBS), fibromyalgia [9], chronic widespread pain [10], depression [11], mood disturbances [12] and insomnia [13]. All of these enigmatic conditions likely share some similar pathology with each other but are often siloed into their own diagnostic landscapes. 

Through the narrative review of recent evidence, this paper proposes the integration of the gut microbiome into—and into alignment with—the diathesis-stress model [8], thereby increasing and expanding its theoretical application to complex musculoskeletal conditions, including chronic musculoskeletal pain conditions. The paper is split into sections for the provision of background for this proposition, followed by presentation of the updated model.

## 2. Background for the Proposed Model

A healthy microbiome is generally defined by considerable biodiversity. Increased numbers and a greater diversity of commensal bacteria are immediately beneficial to the host due to simply outcompeting pathogens for resources. A healthy adult human intestinal tract has about one-hundred trillion microorganisms [14], mainly including—but not limited to—bacteria, fungi, viruses and protozoa, suggesting that our physical composition is as much bacterial as it is human [15,16]. The human microbiome is the name given to the entirety of these microorganisms. Accordingly, humans are “superorganisms”, whereby functional abilities are the result of both innate human physiology and the complimentary or detrimental activity of associated microbes [17]. 

The microbiome impacts on the normal development and lifelong maintenance of immune, neuroendocrine, behavioural and emotional pathways [18,19,20,21]. As such, host–microbial interactions are very likely to be involved in a range of functions, including motor, sensory, emotional, neurotransmission [18,19], endocrinologic [20,21] and immunologic systems [21], and in acute and chronic pain experiences [22].

For example, an abundant and diverse intestinal bacterial population regulates the local regulatory T cell population, preventing or reducing the intestinal inflammation common to Crohn’s disease and inflammatory bowel disease (IBD) [23,24]. However, the richness of the gut microbiome—or lack thereof—is not unique to disorders associated with the digestive tract alone, as it has also been implicated in the pathogenesis of MSK disorders. For example, the inflammatory mechanism in individuals with new-onset rheumatoid arthritis [25], and in individuals without clinical presentation but with systemic autoimmunity associated with rheumatoid arthritis [26], is thought to be perpetuated by overabundance of species called Prevotella copri. This species has been linked to metabolic changes in the gut microbiome, leading to reduce interleukin-18 (IL-18) production and amplified intestinal inflammatory response [27].

An abnormal change in the composition and diversity of commensal resident microbes, referred to as dysbiosis, is considered to be central to the increase in gut permeability which allows pathogens to pass through the epithelial barrier of the gastrointestinal tract, resulting in immune cell activation, a cascade of pro-inflammatory cytokine release, and systemic inflammation [26,28]. Dysbiosis can be triggered by multiple factors, such as alterations in food habits, particularly an increase in the consumption of sugar and proteins, and environmental stressors. A growing body of empirical evidence also suggests that early-life adversities and prolonged exposure to stressors are important determinants for a range of poor physical and mental health conditions in adult life [20,29,30], including pain, via the altered gut microbiome or dysbiosis [31,32]. 

Research also suggests that more children born to mothers with IBD [33] and exposed to higher adverse childhood experiences [34] exhibit dysbiosis in comparison to children born to mothers without these exposures. Stress parameters such as negative emotions, hair cortisol, parasympathetic activity, happiness, and parent-reported emotional problems are also observed to be independently related to gut microbial composition, irrespective of diet [35], in children and adolescents. These observations suggest that dysbiosis—i.e., abnormal alterations in the gut microbial composition—plays an important role in chronic stress-mediated disease vulnerabilities such as inflammation.

## 3. Updated Model

The diathesis-stress model [8] recognized that people bring their unique diatheses (vulnerabilities and protections) to the experience of trauma, positioning these as pre-existing drivers of the peri- and post-trauma experience. In a bio-psycho-socio-spiritual context, Walton and Elliott [8] posited that those individual diatheses play an important role in directing, but not predetermining, a person’s recovery trajectory. We believe that the gut microbiome is an important contributor of the stress-diathesis response, in that dysbiosis is likely to be present prior to the experience of new trauma, and is sufficient to precipitate an adverse chain of events following injury, potentially exacerbated in people with biological and psychological vulnerabilities. Below, we present a narrative overview of recent advances in this field, including potential mechanistic pathways to explain such a link. Drawing on current interdisciplinary research, an integrated conceptual model of the relationship between individual diatheses and the gut microbiome is proposed, with a view to exploring non-recovery in post-injury events (see Figure 1). This updated model explores the complex communication system in the microbiome-gut-brain axis [36,37,38]—mediated via the hypothalamic–pituitary adrenal axis, immune response and cortical functioning [37,38,39,40,41,42,43,44]—and how it may underlie pain and (non-)recovery following the experience of trauma. All three pathways are explored below. 

### 3.1. Hypothalamic-Pituitary-Adrenal Axis 

As a part of the limbic system, the Hypothalamic Pituitary Adrenal (HPA) axis coordinates systemic responses to emotional and biologic stressors [45], largely through the release and regulation of the human stress hormone cortisol. Stress, including physical or emotional trauma, stimulates the secretion of corticotropin releasing factor (CRF) from the hypothalamus, activating the secretion of adrenocorticotropic hormone (ACTH) from the pituitary gland, in turn stimulating the release of cortisol from the adrenal glands, which—under normal conditions—stops the cycle through a negative feedback loop back to the hypothalamus. Cortisol secretion is critical for adapting and responding to adversity, and has varied and wide-reaching systemic effects on the central nervous system (CNS), the immune system, the enteric nervous system (ENS), and the gut mucosa. Emerging evidence points to bidirectional communication between the gut microbiome and the HPA axis. Research shows that the dysbiosis of the gut microbiome can disrupt the functioning of the HPA axis via increased epithelial permeability, the activation of the mucosal innate immune response, and the sensitization of nociceptive sensory pathways, inducing visceral pain [46]. Similarly, chronic low-grade inflammation associated with IBS and depression [47] is believed to be mediated by dysbiosis and bacterial translocation. Furthermore, persistent states of peripheral inflammation through heightened stress biology and/or inflammation can drive deficits in emotional regulation and executive control strategies, enhancing risky behaviours (e.g., a high-fat diet and substance abuse) [48,49,50]. 

This in turn may generate peripheral inflammation due to heightened stress responses [51]. Such biological processes are known to increase engagement in high-risk and unhealthy behaviours (e.g., substance abuse (alcohol, pain medications, a high-fat diet)) to manage the distress [49], and may explain some of the adverse outcomes following the uncertainty introduced by a trauma [52,53,54]. Importantly, network traffic between the brain and the immune system is a two-way street, and inflammatory mediators have been shown to access the brain through multiple mechanisms (e.g., active transport and leaky regions of the blood–brain barrier) [48]. Inflammatory mediators have been shown to alter prefrontal processes, which can further weaken the regulatory influence of the central executive network and the emotional regulation network [49]. This is also the theoretical foundation for our stress-diathesis model of persistent pain development [9]. This model is predicated upon the potential adverse influences of stress dysregulation in the peri- and post-trauma periods, and this framework provides a potential explanatory pathway (through HPAA activity).

It is likely that in the event of trauma/injury, through similar mechanisms, the gut microbiome creates a defaulting reciprocal communication loop that includes the HPA axis, central nervous system (CNS), autonomic nervous system (ANS), and enteric nervous system (ENS). 

### 3.2. Immune Response

A single-cell layer lines the intestine, known as the intestinal epithelial barrier (IEB). These cells are connected via protein complexes, providing a physical barrier between the contents of the intestine and the intestinal wall itself [55]. In healthy intestines, the bacteria rest on the extracellular mucosal layer that covers the apical surface of epithelial cells, and represent a chemical barrier between the intestine lumen and the intestinal wall; they do not come into direct contact with the epithelial cells themselves [55,56]. These epithelial cells can produce various antimicrobial peptides, making them a physical and chemical “retaining wall” against invading microbes [57]. The integrity of the IEB, however, can be compromised in situations of physical or psychological stress, leading to direct contact between gut microbiota and epithelial cells. This phenomenon, often referred to as the “leakiness” of the gut, results in an influx of bacteria and other toxins onto the intestinal wall. Research shows that an influx can trigger an immune response [58], including the activation of mast cells and the alteration of intestinal permeability [59,60,61].

Furthermore, microbiota are recognized by specialised receptors found within the extracellular mucosa and on the surface of intestinal epithelial cells known as Pattern Recognition Receptors (PRRs). These receptors are responsible for recognizing specific bacterial characteristics, and different PRRs respond to different characteristics [62]. Dysregulated communication between PRRs and bacteria is thought to be one of the central causes of chronic intestinal inflammation. Alterations in this interaction can lead to sensitized responses to commensal and pathological species of bacteria, which can result in significant shifts in the types of bacteria that are able to colonize the gut [63]. Once detected by a PRR or by a pain-induced increase in gut permeability [59], bacteria can trigger a flood of small, soluble polypeptides and glycoproteins known as cytokines into the intestinal lining [62,64]. Once released, pro-inflammatory cytokines (like Tumor Necrosis Factor alpha (TNF-α), Interleukin-1 beta (IL-1β) and Interleukin-6 (IL-6)) or anti-inflammatory cytokines (such as Interleukin-10 (IL-10)) modulate inflammation, and can regulate key survival processes by the modulation of cellular differentiation and gene transcription [64,65]. These small peptides are such potent messengers that only a small concentration is required to trigger a significant biochemical signalling cascade [66]. These cytokines behave in an interdependent manner, and have been shown to be present in significant concentrations during inflammation and chronic pain [67]; they are often involved in hyperalgesia and allodynia [68,69,70], nociceptive signalling [71,72,73], protracted tissue sensitivity [69,74], and nerve damage and repair [75,76]. It has been reported that cytokine IL-10 [77] counters the inflammatory and nociceptive effects of cytokines TNF-α, IL-1β, and IL-6 [78,79] in multiple animal models of injury, stress and chronic illness. Many of these cytokine mechanisms have been verified in humans [80], as reviewed by DeVon et al. [81] and Hauser et al. [82].

The microbiota play a pivotal role in modulating these increases in stress and immune reactivity. Their association with the epithelial barrier helps to ensure a regular cycle of immune sensing and accommodation to the various microbes in the gut environment [83]. Crouzet and colleagues [84] observed that gut bacteria taken from patients with IBS and transplanted into germ-free mice result in visceral hypersensitivity in experimentally induced colorectal distension. Although there are shared pathways between immune and nervous systems via PRR, other studies suggest that bacterial products themselves can directly stimulate nociceptive signalling [85]. In a series of elegant experiments, Chiu and colleagues showed that a subcutaneous injection of *Staphylococcus aureus* in mice resulted in heat, cold, and mechanical hypersensitivity. This phenomenon appeared to be closely tied to the bacterial load, rather than immune activation, wherein an increase in the bacterial load resulted in greater hypersensitivity [86]. Taken together, these studies suggest that the presence or absence of certain bacterial species may be linked to nociceptive responses and the generation of visceral and peripheral pain. 

### 3.3. Cortical Functioning

In a comparison of conventionally colonized mice (i.e., mice with a normal gut microbiome) versus germ-free mice (i.e., mice raised in a sterile environment with no microbiome), a significant increase in genes related to myelination in the pre-frontal cortex of the germ-free mice is observed [87]. The absence of the gut microbiome resulted in a hypermyelination of pre-frontal cortex neurons that were not found in any of the limbic structures of the brain. This myelination was reversible once the germ-free mice were colonized with conventional bacteria. It was further indicated that these same changes in myelination are also seen in mice that are deprived of social interaction. Although the precise effects of hypermyelination have not been studied in depth, a phenotypic social withdrawal has been shown to occur in other mouse studies [88]. Of interest is the reintegration of social activity in mice in which the myelination of pre-frontal cortex neurons is corrected. These findings illustrate that the microbiota contribute to the structure of the cortex itself, and may explain some emotional and affective behaviours which are characteristic of chronic pain [87]. Accordingly, the microbiota may not only prove essential for the neural structure, but also necessary for appropriate behavioural conditioning and adaptation in fear-learning situations [89]. 

Gut bacteria have proven to be very capable synthesizers of neurotransmitters themselves, including gamma aminobutyric acid (GABA) (from *Lactobacilllus*, *Bifidobacterium*), norepinephrine (from *Escherichia*, *Bacillus* and *Saccharomyces*) and dopamine (from *Bacillus*) [39]. These inhibitory neurotransmitters have previously been found to attenuate action potential propagation through high-threshold nociceptive neurons, and to reduce sensitization at the spinal cord in fibromyalgia [90]. Forsythe and colleagues [91] found that the chronic treatment of mice with *Lactobacillus rhamnosus* (JB-1) stimulated an increase in central GABA production in the cingulate cortex, prelimbic regions, and the hippocampal regions corresponding to a reduction in stress-induced corticosterone production and anxious and depressive behaviours. This mechanism was thought to occur via the stimulation of the vagus nerve, as severing the nerve fully reversed the effects [91]. It is possible that the microbiome can stimulate an anti-inflammatory reflex through vagal afferents to trigger a release of neurotransmitters that help to suppress inflammation and modulate mood and brain function, and which may have a foundational role in guiding cortical development and function in humans.

## 4. Conclusions

The human gut microbiome has the potential to impact the experience of post-trauma musculoskeletal pain and recovery through a combination of adverse neurobiologic, psychiatric, microbiologic and pathophysiologic sequelae. In this manuscript, we presented an updated model combining the existing stress-diathesis model [8] with the gut microbiome to provide potential mechanistic underpinnings of the clinical course of post-traumatic recovery. This model is intended to stimulate novel research into mechanistic pathways of pain and recovery, and to open new avenues for the exploration of therapeutic targets. Readers should note that despite growing work in humans, evidence for direct causation is difficult to acquire in human studies. The pathways described herein remain theoretical at this time. We have not attempted to be exhaustive in our search for evidence, nor have we conducted formal critical appraisal of the literature used to establish these hypotheses. Caution should be exercised when interpreting this information. Nonetheless, evidence in both animals and humans supports the propositions that dysbiosis may be sufficient to increase the susceptibility to poor pain and affective outcomes and delayed recovery in post-trauma MSK conditions, and that the restoration of “eubiosis” is feasible and may offer a novel therapeutic strategy complementing traditional approaches to the management of chronic pain.

## Figures and Tables

**Figure 1 pathogens-11-00716-f001:**
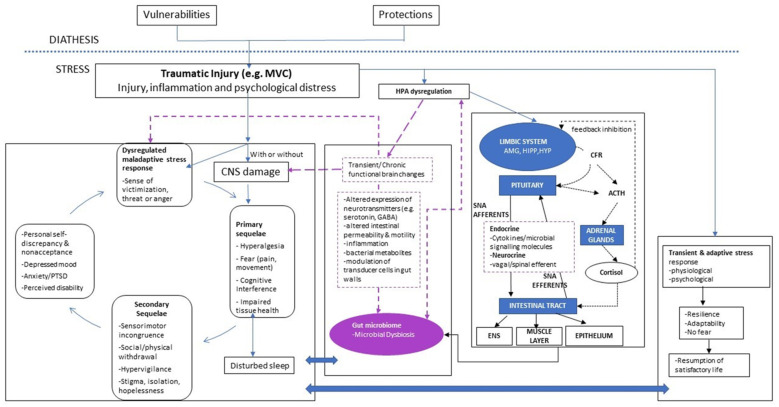
New integrated stress-brain-biome model.

## Data Availability

Not applicable.

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
