# Peer review of "Integrating the Gut Microbiome and Stress-Diathesis to Explore Post-Trauma Recovery: An Updated Model"

_pathogens, 2022, doi:10.3390/pathogens11070716_

Round 1

Reviewer 1 Report

 The manuscript entitled "Integrating gut microbiome and stress-diathesis to explore post-trauma recovery" aims to update and expand the stress-diathesis model by integrating the role of gut microbiome in it and hence, to achieve a better understanding of complex musculosceletal conditions and post-trauma recovery. The paper is based on literature review and does not include specific methodology and research results achieved by the authors. Therefore, it would be more appropriate to be submitted as a review.

 The suggested updated model is strongly motivated by review on the role of gut microbiome in the modulation of HPA axis and immune responses, as well as in cortical functioning. However, the title of the manuscript highlights or suggests specific research on post-trauma recovery based on this model, which is absent in the text. Modification of the title will be beneficial in order to better illustrate the essence of the study.

Specific comments:

11)  L165: revise the sentence. The phrase “on this mucus layer” is not supported by explanation of fact that extracellular mucosal layer covers the apical surface of epithelial cells and represents a chemical barrier between the intestine lumen and the intestinal wall. The physical barrier described in the previous sentence is formed by the complex junctions between epithelial cells and thus, it differs sufficiently from the barrier functions of the mucus layer, which is generated by the same intestinal epithelial cells.

22)    L186-187: differentiation and gene transcription cannot be simply defined as key survival processes. It is better to state: “… can regulate key survival processes by modulation of cellular differentiation and gene transcription”.

33)      L206: Staphylococcus aureus – latin names should be italicized.

Author Response

Reviewer 1

The manuscript entitled "Integrating gut microbiome and stress-diathesis to explore post-trauma recovery" aims to update and expand the stress-diathesis model by integrating the role of gut microbiome in it and hence, to achieve a better understanding of complex musculosceletal conditions and post-trauma recovery. The paper is based on literature review and does not include specific methodology and research results achieved by the authors. Therefore, it would be more appropriate to be submitted as a review.

 The suggested updated model is strongly motivated by review on the role of gut microbiome in the modulation of HPA axis and immune responses, as well as in cortical functioning. However, the title of the manuscript highlights or suggests specific research on post-trauma recovery based on this model, which is absent in the text. Modification of the title will be beneficial in order to better illustrate the essence of the study.

Author response: We thank the reviewer for their feedback. We have revised the title as below. “Integrating gut microbiome and stress-diathesis to explore post-trauma recovery: an updated model”

Specific comments:

1)  L165: revise the sentence. The phrase “on this mucus layer” is not supported by explanation of fact that extracellular mucosal layer covers the apical surface of epithelial cells and represents a chemical barrier between the intestine lumen and the intestinal wall. The physical barrier described in the previous sentence is formed by the complex junctions between epithelial cells and thus, it differs sufficiently from the barrier functions of the mucus layer, which is generated by the same intestinal epithelial cells.

Author response: We thank the reviewer for their feedback. We have now revised this sentence and added an explanation of fact on extracellular mucosal layer (Line 165-166). The paper now reads “In healthy intestines, the bacteria rests on the extracellular mucosal layer that covers the apical surface of epithelial cells and represents a chemical barrier between the intestine lumen and the intestinal wall and does not come in direct contact with the epithelial cells themselves [55,56].

2)    L186-187: differentiation and gene transcription cannot be simply defined as key survival processes. It is better to state: “… can regulate key survival processes by modulation of cellular differentiation and gene transcription”.

Author response: We thank the reviewer for their feedback. We have now revised the definition of gene transcription. (Line 188).

3)      L206: Staphylococcus aureus – latin names should be italicized.

Author response: We thank the reviewer for their feedback. We have italicized Latin names. (Line 208).

Reviewer 2 Report

The manuscript by Manasi Murthy Mittinty et al. presented a new integrated Stress-Brain-Biome model to understand the musculoskeletal conditions of traumatic and non-traumatic origin. Although the idea is relatively new, I do not think it is suitable to be considered for publication as a Research article. Rather, this is just a hypothese. The study is too prelininary and no data was presented to test this hypothesis. I suggest the authors to revise and further discuss this hypothesis. Maybe the manuscript can be resubmitted as an opinion or hypothesis paper. 

Author Response

Reviewer 2

The manuscript by Manasi Murthy Mittinty et al. presented a new integrated Stress-Brain-Biome model

to understand the musculoskeletal conditions of traumatic and non-traumatic origin. Although the idea

is relatively new, I do not think it is suitable to be considered for publication as a Research article.

Rather, this is just a hypothese. The study is too prelininary and no data was presented to test this

hypothesis. I suggest the authors to revise and further discuss this hypothesis. Maybe the manuscript

can be resubmitted as an opinion or hypothesis paper. 

Author response: We thank the reviewer for their feedback. This manuscript presents an updated

model combining the existing stress-diathesis model with the gut microbiome to provide

potential mechanistic underpinnings of the clinical course of post-traumatic recovery. Although

we have not attempted to do a systematic review of literature (we have clearly alluded to this in

the conclusion section) we have conducted formal synthesis of the literature to support the

updated model and discussed in detail. This is an original conceptualization and extension of the

stress-diathesis model and we do believe that it is original in its theory and application. Of

course, with all models, we appreciate they will need updating, revision, if not removal, when

new data becomes available. We believe this is a good step forward in expanding the stress

diathesis model and that new research will include measures captured in our new integrated

Stress-Brain-Biome.

Reviewer 3 Report

In this review, the authors discussed the potential used of gut microbiome in stress-diathesis and post-trauma recovery. The review covers the following points: a) The gut microbiome.  b) The diathesis-stress model. including . Hypothalamic-pituitary-adrenal axis , Immune response, and cortical functioning, 

The idea is not novel, and there are many missing points

a) Animal models supporting the role of gut microbiome in stress diathesis.

b) Role of gut microbiome in other mental disorders having stress diathesis. This point can also beneficial in this review.

c) Any patients derived data showing the success rate. If the author can provide a table showing different cohort studies, this will be great.

d) Another point: it is a review or point of view. It is not an original article. Please verify this in the revised manuscript

Author Response

Reviewer 3

In this review, the authors discussed the potential used of gut microbiome in stress-diathesis and post-trauma recovery. The review covers the following points: a) The gut microbiome.  b) The diathesis-stress model. including . Hypothalamic-pituitary-adrenal axis , Immune response, and cortical functioning, 

The idea is not novel, and there are many missing points

  1. Animal models supporting the role of gut microbiome in stress diathesis.

Author response: We thank the reviewer for their feedback. We have integrated the critical evidence from animal models supporting the role of gut microbiome in stress diathesis (immune response Lines 194-198; Lines 202-213, cortical functioning 215-230; Lines 234-245 as follows:

Lines 194-198: It is reported that cytokine IL-10 [77] counters the inflammatory and nociceptive effects of cytokines TNF-α, IL-1β, and IL-6 [78,79] in multiple animal models of injury, stress and chronic illness. Many of these cytokine mechanisms have been verified in humans [80], as reviewed by DeVon et al [81] and Hauser et al. [82].

Line 202-2013: Crouzet and colleagues [84] observed that gut bacteria taken from patients with IBS and transplanted into germ-free mice results in visceral hypersensitivity in experimentally induced colorectal distension.  Although there are shared pathways between immune and nervous systems via PRR, other studies suggest that bacterial products themselves are able to directly stimulate nociceptive signalling [85].  In a series of elegant experiments, Chiu and colleagues demonstrated that a subcutaneous injection of Staphylococcus aureus in mice resulted in heat, cold, and mechanical hypersensitivity.  This phenomenon appeared to be closely tied to bacterial load rather than immune activation, where an increase in bacterial load resulted in greater hypersensitivity [86].  Taken together, these studies suggest that the presence or absence of certain bacterial species may be linked to nociceptive responses and the generation of visceral and peripheral pain.     

Line 215-230: In a comparison of conventionally colonized mice (i.e. mice with a normal gut microbiome) versus germ-free mice (i.e. mice raised in a sterile environment with no microbiome), a significant increase in genes related to myelination in the pre-frontal cortex of the germ-free mice are observed [87]. The absence of gut microbiome resulted in a hypermyelination of pre-frontal cortex neurons that were not found in any of the limbic structures of the brain.  This myelination was reversible once germ-free mice were colonized with conventional bacteria.  It was further indicated that these same changes in myelination are also seen in mice that are deprived of social interaction.  Although the precise effects of hypermyelination have not been studied in depth, a phenotypic social withdrawal has shown to occur in other mouse studies [88].   Of interest is the reintegration of social activity in mice where myelination of pre-frontal cortex neurons is corrected.  These findings illustrate that the microbiota contributes to the structure of the cortex itself and may explain some emotional and affective behaviours characteristic of chronic pain [87]. Accordingly, the microbiota may not only prove essential for neural structure, but also necessary for appropriate behavioural conditioning and adaptation in fear-learning situations [89].

Line 234-245: These inhibitory neurotransmitters have previously been found to attenuate action potential propagation through high-threshold nociceptive neurons and reduce sensitization at the spinal cord in fibromyalgia [90]. Forsythe and colleagues [91] found that chronic treatment of mice with Lactobacillus rhamnosus (JB-1) stimulated an increase in central GABA production in the cingulate cortex, prelimbic regions, and the hippocampal regions corresponding to a reduction in stress-induced corticosterone production, anxious and depressive behaviours.  This mechanism was thought to occur via stimulation of the vagus nerve as severing the nerve fully reversed the effects [91]. It is possible that the microbiome is capable of stimulating an anti-inflammatory reflex through vagal afferents to trigger a release of neurotransmitters that help to suppress inflammation and modulate mood and brain function and may have a foundational role in guiding cortical development and function, in humans.

  1. Role of gut microbiome in other mental disorders having stress diathesis. This point can also beneficial in this review.

Author response: We thank the reviewer for their feedback. In this manuscript we present an updated model combining the existing stress-diathesis model with the gut microbiome to provide potential mechanistic underpinnings of the clinical course of post-traumatic recovery. This model is intended to stimulate novel research including its application in mental disorders.

  1. Any patients derived data showing the success rate. If the author can provide a table showing different cohort studies, this will be great.

Author response: We thank the reviewer for their feedback. This manuscript proposes the updated model and as indicated in conclusion does not have any pilot data at this stage.

  1. Another point: it is a review or point of view. It is not an original article. Please verify this in the revised manuscript

Author response: We thank the reviewer for their feedback. This manuscript presents an updated model combining the existing stress-diathesis model with the gut microbiome to provide potential mechanistic underpinnings of the clinical course of post-traumatic recovery. Although we have not attempted to do a systematic review of literature (we have stated this in the conclusion section) we have conducted formal synthesis of the literature to support the update model and discussed it in detail. This is an original conceptualization and extension of the stress-diathesis model and we do believe that it is original in its theory and application.

Round 2

Reviewer 1 Report

The authors have fulfilled my specific reccommendations except for the title of the paper.

I'm convinced that this manuscript should be published as a review article.

Author Response

Author response: We have amended the title on reviewer’s request in the revised manuscript and it reads as follows now:

“Integrating gut microbiome and stress-diathesis to explore post-trauma recovery: an updated model”

Reviewer 2 Report

I still think this manuscript is just a hypothese. I do not think this manuscript is suitable to be published as a research paper. However, if the authors insist, I would suggest the authors to add more relavant data to discuss and illustrate the new model. The authors must further revise the manuscript. 

Author Response

Author response: We thank the reviewer for their suggestion.

However, the purpose of this non-systematic review paper is to describe an expanded and integrated diathesis model with a focus on the gut-brain-biome. The model has been proposed through consultation with experts in the field, extensive review of existing evidence, and novel evidence from the authors’ own research programs. A point of convergence currently exists between the psychological and physiological literature that can further understanding of common, yet enigmatic, conditions (such as low back pain, neck pain, musculoskeletal disorders). An expanded model for the development of chronic pain is presented and described, supported by a narrative review of empirical findings that could be used to orient future research efforts towards more interdisciplinary research, and knowledge translation, in the field.

New to our expanded model is the gut-brain combination.  Some may argue these aren’t distinct entities; gut health and brain functioning likely affect both tissue-based and psychological resilience to trauma.  Others may appropriately opine gut health is too broad a term, encompassing all external influences on the individual.  These are intentionally meant to be broad, stimulating research rather than promoting early adoption, which could stifle innovation.  With time and new knowledge some of these domains may become narrowed, others may arise, and some may need to be removed. The core concept is that trauma does not occur in a vacuum, rather all people encounter new trauma with a pre-existing uniqueness that necessarily influences their reactions, and we believe the gut-brain-biome deserves attention in this expanded diasthesis model.

Reviewer 3 Report

No further concerns

Author Response

Author response: We thank the reviewer for their recommendation. We confirm that we have conducted English language spell check.